# Compressive Sensing Based Radio Tomographic Imaging with Spatial Diversity

**DOI:** 10.3390/s19030439

**Published:** 2019-01-22

**Authors:** Shengxin Xu, Heng Liu, Fei Gao, Zhenghuan Wang

**Affiliations:** School of Information and Electronics, Beijing Institute of Technology, Beijing 100081, China; xusx@bit.edu.cn (S.X.); gaofei@bit.edu.cn (F.G.); wangzhenghuan@bit.edu.cn (Z.W.)

**Keywords:** radio tomographic imaging, spatial diversity, compressive sensing, RSS, indoor localization

## Abstract

Radio tomographic imaging (RTI) has emerged as a promising device-free localization technology for locating the targets with no devices attached. RTI deduces the location information from the reconstructed attenuation image characterizing target-induced spatial loss of radio frequency measurements in the sensing area. In cluttered indoor environments, RF measurements of wireless links are corrupted by multipath effects and thus less robust to achieve a high localization accuracy for RTI. This paper proposes to improve the quality of measurements by using spatial diversity. The key insight is that, with multiple antennae equipped, due to small-scale multipath fading, RF measurement variation of each antenna pair behaves differently. Therefore, spatial diversity can provide more reliable and strong measurements in terms of link quality. Moreover, to estimate the location from the image more precisely and make the image more identifiable, we propose using a new reconstruction regularization linearly combining the sparsity and correlation inherent in the image. The proposed reconstruction method can remarkably reduce the image noise and enhance the imaging accuracy especially in the case of a few available measurements. Indoor experimental results demonstrate that compared to existing RTI improvement methods, our RTI solution can reduce the root-mean-square localization error at least 47% while also improving the imaging performance.

## 1. Introduction

In many scenarios, for example, searching for survivors in the disaster area, rescuing hostages and finding the criminals [1], it is critical to locate the target carrying no devices. Conventional device-free localization (DFL) methods are mostly based on video camera, infrared and radar sensors, which are subject to either low penetration capability or high cost. As an effective radio frequency (RF)-based DFL, radio tomographic imaging (RTI) [2] using received signal strength (RSS) has attracted considerable attention in the past decade since RSS measurements are readily available in most wireless commercial off-the-shelf (COTS) devices. Thus, RTI can be implemented on the existing network without any extra hardware, which offers a cost-effective DFL solution. Moreover, since radio signals can penetrate walls and other non-metallic structures, RTI is able to find the target hiding behind obstacles. Recently, RTI has been successfully applied to through-wall target tracking [3,4], residential monitoring [5,6], roadside surveillance [7], obstacle mapping [8,9] and health care [10,11].

It is well understood that when the target obstructs the line-of-sight (LOS) path of a wireless link, RSS of this link will undergo great loss. Inspired by this feature, Wilson [2] originally formulates RTI linear-model method through capturing the RSS variations of wireless links and determining the localization of the target from an attenuation map with a monitored wireless sensor network. RTI achieves a good performance outdoors where multipath does not dominate the signal propagation. However, due to heavy multipath in indoor environments, the blockage of LOS path does not always result in large attenuation of RSS. In fact, RSS can reduce, remain unchanged or even increase, depending on whether the multipath is constructive or deconstructive to the LOS path of the received signals. The ambiguity of RSS change greatly deteriorates the performance of RTI. To overcome this difficulty, many efforts have been made to enhance the robustness of RSS variation in multipath-rich environments. For example, Kaltiokallio [12] employs frequency diversity by measuring RSS on multiple radio channels, in which the nodes are required to frequently switch from one channel to another. Wei [13] proposes to use electronically switched directional antennae to reduce the influence of multipath, but the specially designed antenna will surely increase the system cost. Bocca [14] observes that optimizing the orientation of the antenna can improve the performance of RTI. However, the process of orientation adjustment is time consuming because of its iterative nature.

In most prior work, the node is equipped with one single antenna. Consequently, when the positions of the transmitting node and receiving node are fixed, the RSS variation keeps unchanged and thus lacks diversity. In this paper, we explore the potential of spatial diversity to combat the multipath and improve the performance of RTI in indoor environments. Specifically, to make use of spatial diversity, each node is equipped with multiple antennae. Since wireless devices equipped with multiple antennae are very common nowadays [15], spatial diversity can be achieved using COTS devices. Compared to [12,14], the node does not require to change the operating channel or antenna orientation, which is more bandwidth and time efficient. Considering a link comprised of a transmitting node and a receiving node, any pair of antennae on both nodes can communicate with each other, and thus the link contains a few of sublinks. Note that the received signals are the superimposition of signals via different propagation paths. Therefore, owing to the minor place difference among antennae, the propagation paths of sublinks vary slightly, which will surprisingly lead to large variation of RSS on different sublinks. Averaging the RSS variations on all sublinks in terms of fade level [16] will yield a more robust RSS loss estimator compared to the single antenna configuration.

Another challenge of RTI is to reconstruct the image based on the noisy measurements. Due to the ill-posed problem, conventional RTI methods mostly utilize $\ell_{2}$-norm regularization (known as Tikhonov) [2,12,14], only considering the correlation property in the attenuation map. Although Tikhonov regularization can achieve acceptable localization accuracy, the image result of it is quite noisy and thus unfavorable to target detection. Actually, the target only occupies a few grid areas (tens of grids) compared to the whole monitored region (usually thousands of grids), which suggests that the image reconstruction problem has enough sparsity in nature. As such, compressive sensing (CS) algorithms for sparse signal recovery will be more preferred and robust to noise if we only focus on the target location. Many available CS-based solutions [17], including the least absolute shrinkage and selection operator (LASSO) and orthogonal matching pursuit [18], have been proven to successfully reconstruct the image. Under the framework of Bayesian statistics, Bayesian compressive sensing (BCS) [19] exploits a priori distribution knowledge of attenuation image to improve the recovery accuracy. However, it requires reasonable assumption of priori distribution and is computationally intensive. It is also reported that the localization accuracy of BCS is less accurate than that of Tikhonov [20], but the advantage of CS over Tikhonov is that the reconstructed image is more cleaner. Few researchers have paid their attention to study the combination of above regularizations for RTI. In this paper, we propose a new reconstruction method whose objective function is a linear combination of $\ell_{1}$-norm and $\ell_{2}$-norm regularization, exploiting both the sparsity and correlation in the image. The proposed reconstruction method achieves satisfactory localization performance while its imaging performance is also comparable. The contributions of this paper are as follows:We propose to exploit spatial diversity by using multiple antennae. Benefiting from it, different behaviors of RF measurement variation of each antenna pair can be captured to produce more robust observations for localization.We propose to utilize both the sparsity and correlation inherent in the attenuation image. By linearly combining the $\ell_{1}$-norm and $\ell_{2}$-norm regularization, our reconstruction method can estimate the target location more precisely from the enhanced image, especially in the case of only a few available measurements. Moreover, the target’s profile also can be kept accurately.Real indoor experiments are conducted to validate the effectiveness of the proposed method. The results show that both the localization accuracy and image quality can be improved by spatial diversity and the proposed reconstruction method.

The rest of the paper is organized as follows. Section 2 gives a review of the related work of RTI. Section 3 briefly describes the problem statement. The spatial diversity method is presented in Section 4 and the new reconstruction method is introduced in Section 5. Section 6 shows the experiment, followed by the results analyzed in Section 7. Section 8 discusses some related issues and Section 9 concludes the paper.

## 2. Related Work

RF-based DFL systems exploit some handy wireless measurements from ubiquitous wireless devices to localize targets without being equipped with any electronic devices or tags. Since there is no need for target cooperation, extensive attention has been attracted on the development of RF-based DFL systems. RF-based DFL can be mainly classified into two categories: fingerprint-based [21] and model-based [2]. Since the former technique requires large time-consuming labor efforts of offline RSS map establishment, we focus our interest on the model-based DFL method, more specifically, radio tomographic imaging (RTI).

In model-based DFL systems, spatial models relating RSS measurements to the target location are needed to be established. Original RTI [2] employs a simple linear model that is based on discretized pixels. Following that, some pixel-free nonlinear models to directly associate the RSS measurement with the target location, such as exponential model [22,23], magnitude model [24] and exponential-rayleigh model [25], are empirically derived through large amount of experimental data fitting. Based on diffraction theory, more complicated analytical models [26,27,28,29] are also developed. Although these models are more precise than the linear model, they are usually computationally intensive and should be combined with sequential Monte Carlo filtering techniques to infer the location information of targets. This paper focuses on the improvement of the linear RTI method since it is more easily to implement for pratical applications.

The ability of RTI methods to realize a high localization accuracy in indoor scenarios is limited by a multipath effect. When multipath propagation dominates the way radio signals are transmitted, target-induced RSS change of LOS of a link will become ambiguous rather than significantly attenuated. Many efforts have already been devoted to reduce this uncertainty. Kaltiokallio [12] exploits channel diversity to improve link quality in frequency domain by averaging the RSS over link-specific strong channels based on fade level criterion. Multi-frequency sub 1-GHz RTI [30,31], which is less susceptible to multipath effects, is investigated to be feasible for large-scale environments. Wei [13] implements a directional RTI to improve localization accuracy by using electronically switched directional antennae instead of omni-directional counterparts. Furthermore, an E-shaped patch antenna is specially designed to achieve 43% accuracy improvement for through-wall RTI [32]. In Ref. [14], RF sensors are iteratively rotated to optimize antenna orientation for best performance. A novel mmRTI using highly-directional 60 GHz sensing networks, is presented in Ref. [33] to locate targets accurately in rich multipath environments. All above improvements are aimed at making the RSS of the LOS path more robust. In this paper, we also achieve this goal using spatial diversity through multiple antennae mounted.

RTI usually uses $\ell_{2}$-regularized methods [34] to reconstruct the attenuation image from which the target’s location can be inferred. Since the target distributes sparsely in the monitored area, CS-based image reconstruction methods can be suitable for RTI, especially when a few measurements are available. Efficient CS algorithms for sparse signal recovery have been extensively surveyed and compared in Refs. [17,35]. Kanso [36] first implements CS-based RTI and presents the performance of LASSO and OMP. A novel Bayesian greedy matching pursuit [37] is proposed for image reconstruction from a small set of measurements. Ref. [38] explores the BCS method to achieve compressive obstacle mapping. By incorporating heterogeneous noise prior models into BCS, heterogeneous BCS [20] is developed to enhance the compressive RTI performance. Although CS-based reconstruction methods enable RTI more efficient and improve the localization accuracy to a certain extent, the target’s profile in the attenuation image would be destroyed. In this paper, we take care both the localization accuracy and image quality through the combination of $\ell_{1}$-norm and $\ell_{2}$-norm regularization, which is expected to allow a high localization accuracy while maintaining a clearly identifiable image.

## 3. Problem Statement

As illustrated in Figure 1, *K* nodes with their positions $\left(x_{k},y_{k}\right)$, k=1,2,…,K known a priori are fixed around the perimeter of the monitored region. In the network, each pair of nodes can comprise a link, leading to $L=K\left(K-1\right)$ bidirectional links in total. In addition, to achieve spatial diversity, each node is equipped with $n_{A}$ antennae which can work independently, shown in Figure 1b. Therefore, each pair of nodes includes $M=n_{A}^{2}$ links, which are called sublinks to avoid confusion. The node can provide the RSS measurement of each sublink. When the target moves in the monitored region, the RSS of links will change due to the diffraction, reflection or scattering of the target. In particular, when the target blocks the wireless links, the RSS of these links will experience large attenuation, which allows us to localize the device-free target.

## 4. Spatial Diversity

The first step of RTI is to measure the RSS variation of all sublinks. For clarity, we denote ${\bar{r}}_{l}^{m}$ as the RSS of the m-th
$\left(m=1,2,\ldots ,M\right)$ sublink of the l-th
$\left(l=1,2,\ldots ,L\right)$ link when the target is absent. According to the path loss model [39], ${\bar{r}}_{l}^{m}$ can be written as
(1)$${\bar{r}}_{l}^{m}=P_{T}-10n_{p}\mathrm{lg}d_{l}^{m}+v_{l}^{m},$$
where $P_{T}$ is transmit power, $n_{p}$ is the path loss exponent, $d_{l}$ is the distance between the two nodes of link *l*, and $v_{l}^{m}$ is the noise due to the multipath fading and shadow fading in the environment.

When the target is in the monitored region, the target will obstruct some wireless links, resulting in a great drop of the RSS of those links. In addition, the target will also affect multipath signals, which also contributes to the RSS variation. Thus, the corresponding RSS can be represented as
(2)$$r_{l,t}^{m}=P_{T}-10n_{p}\mathrm{lg}d_{l}^{m}-S_{l,t}+v_{l}^{m}-n_{l,t}^{m},$$
where $S_{l,t}$ is the RSS loss induced by the blockage of the target and $n_{l,t}^{m}$ is the noise attributing to the target’s influence on multipath. Therefore, by subtracting Equation (Equation 1), we can obtain the RSS variation due to the presence of the target as
(3)$$\Delta r_{l,t}^{m}={\bar{r}}_{l}^{m}-r_{l,t}^{m}=S_{l,t}+n_{l,t}^{m}.$$

We can see that the RSS loss is corrupted by multipath noise. Since the variance of noise closely depends on the number of multipath in the environment, the more cluttered the environment, the larger the noise and vice versa. Moreover, it is well known that the received signal is the phasor sum of the duplicated signals from all propagation paths, meaning that small variation of a propagation path can cause a large change of the RSS. In other words, the RSS variation is sensitive to multipath. Note that, in the multi-antenna configuration, while the propagation paths of the sublinks slightly vary, the RSS variations of sublinks will remarkably differ. That is the foundation on which the spatial diversity is based.

To demonstrate the impact of multipath on the variation of RSS, we collect the RSS variations of a link when it is crossed by a person. In this simple experiment, each node has three antennae, and there are 3×3=9 sublinks for this link. Figure 2 plots RSS measurements of the nine sublinks. We can conclude two important facts from the experiment measurement. First, due to multipath, not all the sublinks’ RSS undergo large attenuation when the target obstructs the link. For example, the RSS of the sublinks 3 and 5 do not change much when the link is blocked, and the RSS of the sublink 4 even increases. Therefore, we can imagine that if the link happens to be one of the sublinks 3, 4 or 5 in the single antenna configuration, it will be difficult to localize the target.

Second, the RSS variation of the sublink is also closely related to the static RSS (i.e., when the link is not blocked) of the sublink itself. Generally, the larger the sublink’s static RSS, the more preferable RSS variation obtained when the sublink is blocked by the target. For example, as shown in Figure 2, the RSS of any sublink experiencing large shadowing loss is larger than −50 dBm. To explain and exploit this feature, Refs. [40,41] propose the concept of fade level which is a function of RSS to evaluate the link quality. The fade level of a sublink in our paper is defined as
(4)$$F_{l}^{m}={\bar{r}}_{l}^{m}-\min_{m}\;{\bar{r}}_{l}^{m}.$$

From the definition of the fade level, we can see that the larger RSS of the sublink, the higher its fade level. To reduce the impact of noise on the RSS variation, we propose to weight RSS variations of the sublinks in terms of the fade level, which yields
(5)$$\Delta r_{l,t}=\frac{1}{\sum_{m=1}^{M}F_{l}^{m}}\sum_{m=1}^{M}F_{l}^{m}\Delta r_{l,t}^{m}=S_{l,t}+n_{l,t},$$
where $n_{l,t}=\frac{1}{\sum_{m=1}^{M}F_{l}^{m}}\sum_{m=1}^{M}F_{l}^{m}n_{l,t}^{m}$.

## 5. Image Reconstruction

The following step of RTI is to reconstruct the image based on RSS variations. RTI assumes that the RSS loss is a spatial integration of the value occurred in the propagation field of the monitored region. Thus, if the monitored area is uniformly divided into grids, the integration can be written as [2]
(6)$$\Delta r_{l,t}=\sum_{j=1}^{N}w_{l,j}\Delta x_{j,t}+n_{l,t},$$
where $\Delta x_{j,t}$ denotes the RSS loss occurred in the j-th grid and $w_{l,j}$ is the weight of the j-th grid with respect to the l-th link and *N* is the number of grids. Intuitively, since radio signals mainly propagate along the LOS path in the absence of multipath, the closer the grid is to the LOS path, the larger weight should be assigned to the grid. To model this fact, a spatial elliptical model is proposed in Ref. [42], in which the weight can be approximately calculated as
(7)$$w_{l,j}=\left\{\begin{array}{c} 1,d_{l,j}\left(1\right)+d_{l,j}\left(2\right)<d_{l}+\lambda ,\\ 0,\;\mathrm{otherwise}, \end{array}\right.$$
where $d_{l,j}\left(1\right)$ and $d_{l,j}\left(2\right)$ are the distances between the j-th grid and the two nodes comprising the link *l*, respectively, and λ is a tunable parameter controlling the width of the ellipse. Actually, in accordance with Fresnel diffraction theory, the attenuation primarily occurs within the first Fresnel zone of the link [39]. As a result, the parameter λ can be chosen as one half of the wavelength, which is equal to 0.0625 m if the nodes operate at a 2.4 GHz band.

If we take into consideration *L* links within the monitored region, Equation (Equation 6) can be rewritten as
(8)$$\Delta r_{t}=Wx_{t}+n_{t},$$
where the RSS variation vector $\Delta r_{t}={\left[\Delta r_{1,t},\Delta r_{2,t},\ldots .,\Delta r_{L,t}\right]}^{T}\in R^{L}$, the noise vector $n_{t}={\left[n_{1,t},n_{2,t},\ldots .,n_{L,t}\right]}^{T}\in R^{L}$, the RSS attenuation vector to be estimated $x_{t}={\left[\Delta x_{1,t},\Delta x_{2,t},\ldots .,\Delta x_{N,t}\right]}^{T}\in R^{N}$, the weight matrix $W={\left[w_{1},w_{2},\ldots ,w_{L}\right]}^{T}\in R^{L\times N}$, $w_{l}={\left[w_{l1},w_{l2},\ldots ,w_{lN}\right]}^{T}\in R^{N}$.

Since the weight matrix W is rank deficiency (L≪N), regularization should be imposed to make the solution stable. In the original form of RTI [2], Tikhonov regularization is inserted, whose objective function takes the form of
(9)$$\min_{x_{t}}\;G\left(x_{t}\right)={∥\Delta r_{t}-Wx_{t}∥}_{2}^{2}+\mu_{2}x_{t}^{T}C^{-1}x_{t},$$
where $C\in R^{N\times N}$ is a prior covariance matrix of $x_{t}$ and $\mu_{2}$ is the $\ell_{2}$-norm regularization parameter. The prior knowledge of $x_{t}$ is a reasonable assumption that the distribution of the attenuation map is a Gaussian process [12,14], an exponential decaying function is employed to approximately calculate the covariance matrix, i.e.,
(10)$${\left[C\right]}_{ij}\approx \exp\left(-\frac{d_{ij}}{\delta }\right),$$
where $d_{ij}$ is the Euclidean distance between the i-th grid and the j-th grid, and δ is the decaying parameter.

The imaging result of Tikhonov regularization is usually blurred with the noise. In fact, the target occupies only a little space compared to the whole monitored area, suggesting that $x_{t}$ is sparse in nature. Using the sparsity can further reduce the impact of the noise. CS is a popular method in the past decade to recover the signal owning sparse property, in which $\ell_{1}$-norm of the signal is incorporated. For example, Kanso [36] proposes the LASSO method, which solves
(11)$$\min_{x_{t}}\;G\left(x_{t}\right)={∥\Delta r_{t}-Wx_{t}∥}_{2}^{2}+\mu_{1}{∥x_{t}∥}_{1},$$
where ${∥x_{t}∥}_{1}$ is $\ell_{1}$-norm of $x_{t}$ defined as ${∥x_{t}∥}_{1}=\sum_{j=1}^{N}\left|x_{j,t}\right|$. Equation (Equation 11) does not consider the correlation of $x_{t}$, which will increase the localization error. Therefore, we propose a new image reconstruction method considering both the correlation and sparsity of $x_{t}$, and hence the objective function can be rewritten as
(12)$$\min_{x_{t}}\;G\left(x_{t}\right)={∥\Delta r_{t}-Wx_{t}∥}_{2}^{2}+\mu_{2}x_{t}^{T}C^{-1}x_{t}+\mu_{1}{∥x_{t}∥}_{1},$$
where $\mu_{1}$ is $\ell_{1}$-norm regularization parameter.

Sorting Equation (Equation 12), we have
(13)$$\begin{array}{cc} G\left(x_{t}\right)= & x_{t}^{T}\left(W^{T}W+\mu_{2}C^{-1}\right)x_{t}-2x_{t}^{T}W^{T}\Delta r_{t}\\ & +\;\mu_{1}{∥x_{t}∥}_{1}+\Delta r_{t}^{T}\Delta r_{t}. \end{array}$$
Since $W^{T}W+\mu_{2}C^{-1}$ is positive definite, the Cholesky decomposition is
(14)$$W^{T}W+\mu_{2}C^{-1}=Q^{T}Q,$$
where $Q\in R^{N\times N}$. Substituting Equation (Equation 14) into Equation (Equation 13), the objective function becomes
(15)$$\begin{array}{cc} G\left(x_{t}\right) & =x_{t}^{T}Q^{T}Qx_{t}-2x_{t}^{T}W^{T}\Delta r_{t}+\mu_{1}{∥x_{t}∥}_{1}+\Delta r_{t}^{T}\Delta r_{t}\\ & =x_{t}^{T}Q^{T}Qx_{t}-2x_{t}^{T}Q^{T}{\left(Q^{T}\right)}^{-1}W^{T}\Delta r_{t}+\mu_{1}{∥x_{t}∥}_{1}\\ & \;\;\;\;\;+\Delta r_{t}^{T}\Delta r_{t}\\ & ={∥b_{t}-Qx_{t}∥}_{2}^{2}+\mu_{1}{∥x_{t}∥}_{1}-b_{t}^{T}b_{t}+\Delta r_{t}^{T}\Delta r_{t}\\ & ={∥b_{t}-Qx_{t}∥}_{2}^{2}+\mu_{1}{∥x_{t}∥}_{1}+C, \end{array}$$
where $b_{t}={\left(Q^{T}\right)}^{-1}W^{T}\Delta r_{t}\in R^{N}$ and $C=\Delta r_{t}^{T}\Delta r_{t}-b_{t}^{T}b_{t}$ are constants independent of $x_{t}$. Comparing Equation (Equation 15) with Equation (Equation 11), we can see that the objective function has been transformed to the standard form of LASSO method. The above objective function is convex, meaning that it can be efficiently solved by convex programming. At present, there are already some powerful solvers for this type of problem, for example, l1-ls [43], CVX [44,45] and so on.

When there is only one target to be located, the largest entry in $x_{t}$ reveals the position of the target, i.e.,
(16)$$I_{t}=\arg\max_{j}\;x_{j,t}.$$
Thus, the coordinate of the grid $I_{t}$ can be viewed as the position estimation of the target. When considering multiple targets in the monitored region, the targets will correspond to different blobs in the reconstructed image, from which the localization of multiple targets can be accomplished by clustering techniques, for example, *k*-means clustering or hierarchical agglomerative clustering (HAC) [41]. The center of the clusters can be estimated as the position of the targets.

## 6. Experimental Validation

In this section, we describe the details of experimental settings and present numerical metrics for performance evaluation.

### 6.1. Experimental Setup

We conduct experiments in a typical conference room as shown in Figure 3a, which is furnished with a large number of objects including desks, chairs and concrete walls. The monitored region covering an area of $4.8\;m\times 2.4\;m=11.52\;m^{2}$ is surrounded by 12 stands with a height of 0.9m, evenly spaced by 1.2 m. Wireless nodes we employed are TI CC2530 half-duplex radios, which use the 2.4 GHz ISM frequency band and omni-antennae. Three nodes are placed on each stand in order to achieve spatial diversity. Therefore, in this configuration, we have a total of 12×11=132 direct links and 3×3=9 sublinks per link. We pick up 21 test positions uniformly distributed within the monitored region, as shown in Figure 3b. Nodes are programmed to run a single-channel token passing communication protocol. At each test position, a person stand still for about 15 s. RSS measurements of a link are averaged over time to reduce the impact of measurement noise.

### 6.2. Numerical Metrics

We introduce two metrics to evaluate the performance of our proposed method.

For localization accuracy, We use root mean square error (RMSE) to assess the localization performance, which is defined as
(17)$$\epsilon_{l}=\sqrt{\frac{1}{N_{T}}\sum_{k=1}^{N_{T}}{∥z_{k}-{\hat{z}}_{k}∥}^{2}},$$
where $N_{T}$ is the number of test positions and $∥z_{k}-{\hat{z}}_{k}∥$ is the Euclidean distance between the true coordinate $z_{k}$ and estimated coordinate ${\hat{z}}_{k}$ of the k-th test position.

For image quality, we adopt the mean square error (MSE) of an image which quantifies the intuitive dissimilarity between the reconstructed image ${\hat{x}}_{r}$ and the true image $x_{t}$, which is calculated by
(18)$$\sigma_{i}=10\log_{10}\left(\frac{{∥x_{t}-{\hat{x}}_{r}∥}^{2}}{N}\right).$$

Note that both image values are scaled in the range [0,1]. RMSE of all the image of test positions is expressed as
(19)$$\epsilon_{i}=\sqrt{\frac{1}{N_{T}}\sum_{k=1}^{N_{T}}\sigma_{i}^{2}}.$$

The true image $x_{t}$ can be obtained by modeling the cross section of the person as a rectangle of 40 cm by 20 cm, which is given by [2]
(20)$${\left[x_{t}\right]}_{j}=\left\{\begin{array}{cc} 1, & \mathrm{if}\;\mathrm{the}\;\mathrm{person}\;\mathrm{occupies}\;\mathrm{grid}\;j,\\ 0, & \mathrm{otherwise}. \end{array}\right.$$

## 7. Performance Analysis

In this section, we evaluate the effectiveness of our proposed RTI method with experimental data, and compare it against the traditional RTI [2], channel diversity RTI [12], and rotating RTI [14] in terms of localization accuracy and image quality. For simplicity of description, all the mentioned RTI algorithms are abbreviated as SD-RTI, RTI, CD-RTI, and RO-RTI in respective order. In addition, we also present performance comparison between our proposed integrated image reconstruction solution ($\ell_{1}+\ell_{2}$) and the $\ell_{2}$-norm solution of Tikhonov regularization and the $\ell_{1}$-norm solution of LASSO method. In the simulation, the size of the grid is 0.1m×0.1m, the decaying parameter is set to δ=10, the regularization parameters for $\ell_{1}$-norm term and $\ell_{2}$-norm term are set to $\mu_{1}=10$, $\mu_{2}=1$, respectively. All the image reconstruction parameters are optimized by the grid search method.

First, we study the advantage of the use of spatial diversity. One additional experiment is performed with placing one node at each stand for original RTI and CD-RTI. A multi-channel protocol with a list of eight specified channels ({11,13,15,17,19,21,23,25}) with 10 MHz apart is used for CD-RTI, and measurements of channel 11 are selected for original RTI. We optimize antenna orientation of one link for RO-RTI by finding the strongest sublink among the total of nine sublinks. For all the RTI algorithms to be compared, the proposed regularization method is applied for location determination and image reconstruction. Table 1 tabulates the RMSE, median and standard deviation of localization and imaging results with different RTI algorithms. Figure 4 also shows the localization error at individual test positions. Compared to the original RTI, the localization accuracy and image quality are both significantly improved by using spatial diversity. From Figure 4, we can see that, without spatial diversity, the localization error at some positions (e.g.,L6, L7) can be up to 2 m. The poor performance achieved is primarily due to serious multipath fading at these positions. On the other hand, there is no large localization error for SD-RTI, which confirms that the RSS variation is less sensitive to multipath after weighted averaging the RSS variation of sublinks with respect to fade level. While other improvements with channel diversity (CD-RTI) and orientation optimization (SD-RTI) estimate the target location more accurately, SD-RTI also performs better than them. On average, SD-RTI improves the localization accuracy by 88% (compared to RTI), 55% (compared to CD-RTI) and 47% (compared to RO-RTI), respectively. As for image quality, using spatial diversity reconstructs the best attenuation image, approximately 5 dB lower than the original RTI in terms of RMSE. Although the promotion is not very substantial compared against CD-RTI (5%) and RO-RTI (6%), the deviation of imaging accuracy with SD-RTI is smallest, achieving a higher overall imaging performance.

Next, we evaluate the performance of our proposed reconstruction method. RSS measurements we used for all the image reconstruction methods are provided by SD-RTI. Table 2 lists localization and imaging results of Tikhonov, LASSO and proposed reconstruction methods, respectively. From Table 2, we can observe that the Tikhonov method achieves the satisfactory localization accuracy but with the largest imaging error. On the contrary, the LASSO method gives the best image performance but has a larger localization error than that of the Tiknohov method. Combining the advantages of both, the proposed reconstruction method gets the best localization performance while image quality of it is also comparable to that of the LASSO method (only reduced by 2%). As an example of illustration, Figure 5 shows the image of three reconstruction methods when the target is located at a certain position L16 (3.6 m, 0.6 m). We can see that the image of $\ell_{2}$ regularization is blurred and corrupted by noise. Even worse, due to multipath effects, there is a large bright part at the top edge of the image, originated from unexpected RSS changes of some links not blocked by the target. By contrast, using the LASSO and proposed method, the image is reconstructed to be more clean and includes no much noise. More importantly, our proposed method also keeps accurately the profile of the target while the other two methods fail to capture that. This enhancement is helpful for the applications of target detection and interest of area mapping.

## 8. Discussion

In this section, we discuss the effect of the degree of spatial diversity on the performance and system cost of our proposed method. Moreover, future promising research directions are also presented.

### 8.1. Impact of Number of Sublinks

We consider the number of sublinks as the degree of spatial diversity. How the localization and imaging performance are influenced by the used number of sublinks is shown in Figure 6. We traverse all combinations of 9m sublinks for m∈{1,2,⋯,9} and then calculate the average RMSE over all combinations for *m*. When m>4, the performance improvement tends to be smooth, which means that a few number of sublinks is enough to enjoy the benefits of space diversity. In some application scenarios where there are lack of sufficient informative sublinks, the performance of RTI can be enhanced by this advantage. As shown in Figure 6a, the localization performance gap between the proposed regularization and Tikhonov shrinks down with the increasing number of sublinks. In the case of only a few sublinks available, the localization accuracy of the proposed method is much better than that of Tikhonov. The reason is that the proposed reconstruction method utilizes the sparsity of $x_{t}$, excludes some unexpected interference and thus is robust to noise, especially for the case of the small number of sublinks. When the number of sublinks is more than six, RMSEs of the two methods are almost the same. However, Figure 6b illustrates that the image quality of the proposed reconstruction is always much higher than that of Tikhonov, achieving an enhancement of about 7 dB regardless of the number of sublinks. Moreover, from Figure 6, we can see that, exploiting the correlation property of $x_{t}$, the proposed reconstruction method can locate the target more accurately than LASSO regularization while image quality of them is very close. As the number of sublinks increases, performance gaps between them become gradually narrowing. In summary, compared to Tikhonov and LASSO, the proposed reconstruction method makes a good compromise between the localization and imaging performance.

### 8.2. System Cost

From above analysis, we can know that performance gains from our spatial diversity are at the expense of multiple antennae used. In this paper, we emulate it through multiple nodes arranged in place, which can be regarded as one node with multiple antennae. It may be worried that this system will require more nodes so as to achieve desired accuracy. This can increase the system cost and make it impractical. However, there is no need to concern this problem. First, we have already demonstrated that the number of sublinks required is not too large for better performance. Second, nowadays our surroundings are full of ubiquitous WiFi devices, which are generally equipped with three or more antennae. Therefore, our system can be easily implemented in commodity devices and poses no additional cost. Finally, our system only employs the multiple antenna configuration and can avoid shortcomings of frequent channel switching procedure in CD-RTI and intractable antenna orientation adjustment process in RO-RTI. In addition, computational complexity of our proposed reconstructed method is comparable to LASSO because of the same equivalent form.

### 8.3. Future Work

RTI usually leverages RSS as RF measurements to realize target localization. RSS is a coarse indicator of signal feature quantity obtained from the MAC layer, which characterizes the total power of synthesized signal. When we implement our system with existing WiFi devices, with the help of some powerful tools, we can extract extensive informative information describing channel characteristics in physical layer between multiple antennae, i.e., fine-grained CSI samples including magnitude and phase [15]. Currently, CSI is mostly used in fingerprint-based DFL. Therefore, one promising direction of research is to examine the ability of CSI for RTI. We believe that CSI-based RTI will outperform RSS-based RTI, since CSI (1) is more robust to multipath fading; (2) supports the channel diversity and spatial diversity simultaneously; and (3) provides raw information containing angle of arrival to be further studied.

Another challenging problem in RTI research is the localization and tracking of more than one target. Since multi-target-induced shadowing is more difficult to model theoretically than one target, most research focused on multiple targets localization assume that RSS variation observed is simply the superimposition of that of single-target and requires some clustering algorithms to distinguish them separately. In future research, we will investigate how the correlated effect of multiple targets is reflected on RF measurements of multiple antennae and seek to achieve accurate multi-target RTI localization using a minor path difference caused by spatial diversity.

## 9. Conclusions

In this paper, we have presented some improvements to the localization and imaging performance of RTI in challenging environments. We first propose using spatial diversity for RSS attenuation modeling and then explore the combination of $\ell_{1}$-norm and $\ell_{2}$-norm regularization for image reconstruction. In spatial diversity, due to the minor position difference of the antennae, RSS variations on the sublinks caused by the target significantly vary. A more robust RSS variation estimator can be obtained through averaging the RSS variations on the sublinks with respect to link quality level. To enhance the attenuation image quality, by incorporating the sparseness and correlation implied in the image, our new reconstruction method can preserve the cross section outline of the target clearly as well as achieve accurate localization estimation. This multiple antenna configuration can be easily deployed in real applications with commodity devices. We also have shown that spatial diversity is more suitable for cases where a few measurements are available. Experimental results confirm that the localization accuracy and image quality can be further improved with spatial diversity and the proposed reconstruction method compared to existing RTI methods. 

## Figures and Tables

**Figure 1 sensors-19-00439-f001:**
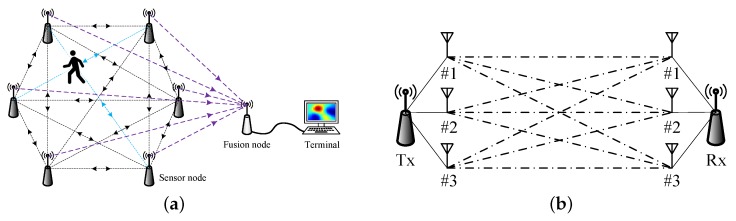
The proposed RTI localization scheme: (**a**) a typical RTI network illustration; (**b**) each wireless link consists of multiple sublinks. In (**a**), the double arrow indicates the bidirectional link and the blue color represents the links obstructed by the target. In (**b**), each node is equipped with three antennae, resulting in a link with nine sublinks.

**Figure 2 sensors-19-00439-f002:**
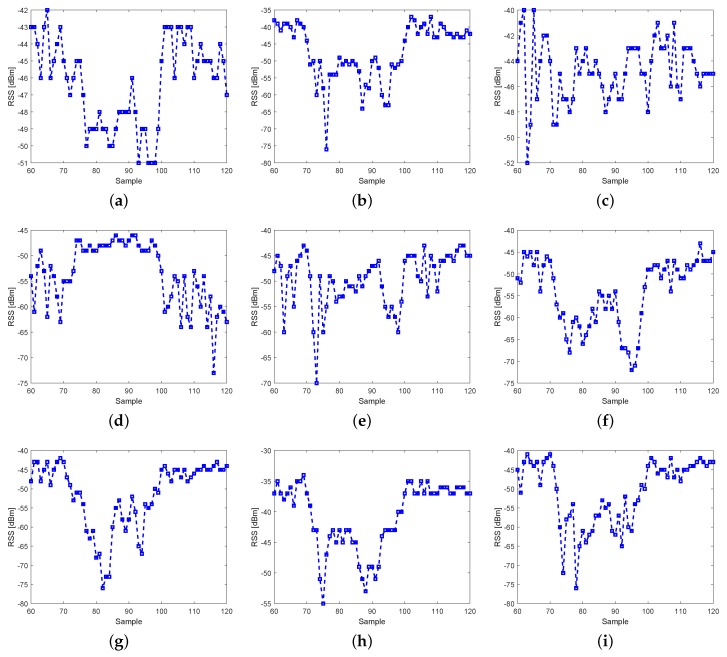
Temporal variation of RSS measured on nine sublinks of the link between node 6 and node 9 with human movement. (**a**) sublink #1; (**b**) sublink #2; (**c**) sublink #3; (**d**) sublink #4; (**e**) sublink #5; (**f**) sublink #6; (**g**) sublink #7; (**h**) sublink #8; (**i**) sublink #9.

**Figure 3 sensors-19-00439-f003:**
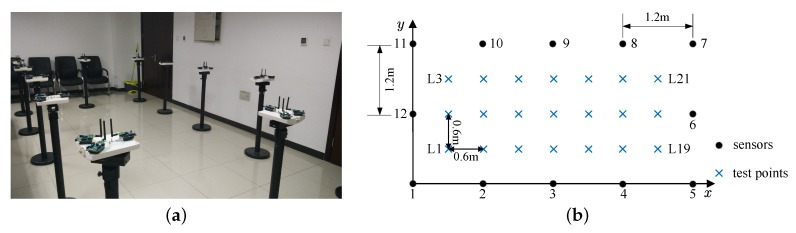
Conference room: (**a**) photography; (**b**) layout. In the test, the orientation of the target is perpendicular to the *y*-axis.

**Figure 4 sensors-19-00439-f004:**
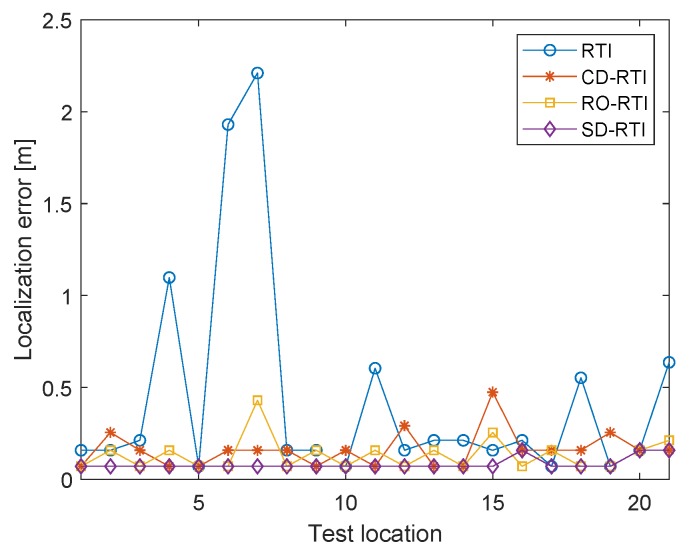
Localization error at individual test locations. SD-RTI achieves the best localization results.

**Figure 5 sensors-19-00439-f005:**
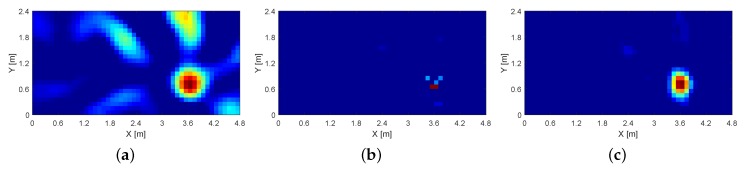
The attenuation image reconstructed using (**a**) Tikhonov ($\ell_{2}$-norm); (**b**) LASSO ($\ell_{1}$-norm); and (**c**) proposed ($\ell_{1}+\ell_{2}$). The image quality of the proposed method is more clear and accurate than the other two methods.

**Figure 6 sensors-19-00439-f006:**
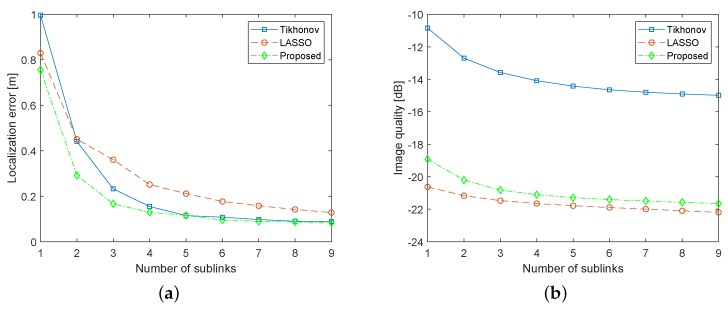
Impact of the number of sublinks on (**a**) localization accuracy; and (**b**) image quality.

**Table 1 sensors-19-00439-t001:** Performance evaluation indicators of different RTI algorithms.

		RTI	CD-RTI	RO-RTI	SD-RTI
Localization Accuracy [m]	RMSE	0.7324(88%)	0.1858(55%)	0.1581(47%)	0.0831
	Median	0.1581	0.1581	0.0707	0.0707
	Standard Deviation	0.5989	0.0975	0.0885	0.0263
Image Quality [dB]	RMSE	−16.9590(27%)	−20.3113(6%)	−20.5741(5%)	−21.6545
	Median	−17.5586	−20.5984	−20.9777	−21.8022
	Standard Deviation	2.9471	2.3440	1.3634	0.6691

**Table 2 sensors-19-00439-t002:** Performance evaluation indicators of different regularization algorithms.

		Tikhonov	LASSO	Proposed
Localization Accuracy [m]	RMSE	0.0886(6%)	0.1282(35%)	0.0831
	Median	0.0707	0.0707	0.0707
	Standard Deviation	0.0313	0.0664	0.0263
Image Quality [dB]	RMSE	−14.9858(44%)	−22.1956(−2%)	−21.6545
	Median	−14.9986	−22.0116	−21.8022
	Standard Deviation	1.2716	0.9476	0.6691

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
