# Peer review of "Compressive Sensing Based Radio Tomographic Imaging with Spatial Diversity"

_sensors, 2019, doi:10.3390/s19030439_

Reviewer 1 Report

The authors propose to enhance the performance of tomography imaging by using the spatial diversity and compressive sensing to reconstruct less noisy images. My comments are as follows:

1.     The authors assume that the wireless node is equipped with multiple antennas to use the spatial diversity and argue that most of prior works considered one single antenna. The proposed method thus proposes a change in the structure of the wireless node which is an attractive solution. This solution is costly in terms of hardware. How much improvement can we get using the new approach? and is this improvement worthy compared with the additional cost of proposed method. The authors should discuss these points to build a strong argument.

2.     The paper is readable, but the authors need to improve the quality of the writing and the presentation. There are some typos and grammatical errors that need to be fixed.

3.     In the abstract, the authors state that “The results show that the performance of RTI can be significantly improved by spatial diversity and proposed compressive sensing”. How much improvement? and compared to what?

4.     The authors proposed a new recovery algorithm to reconstruct the image from noisy measurements. The proposed algorithm is compared to some techniques. Under which conditions the proposed recovery method is successful? How many measurements are required and what is the relationship between the sparsity and the measurement number used? Why the authors need a new recovery algorithm if there are numerous algorithms: Bayesian method, greedy techniques, and convex/relaxation techniques that have been shown they can achieve a high reconstruction accuracy even from a noisy measurement.

5.     Some recent works on RIT and compressive sensing that should be considered.

o    Thu L. N. Nguyen and Yoan Shin, “Deterministic Sensing Matrices in Compressive Sensing: A Survey,” The Scientific World Journal, vol. 2013, Article ID 192795, 6 pages, 2013. https://doi.org/10.1155/2013/192795.

o   Y. Arjoune, N. Kaabouch, H. El Ghazi, and A. Tamtaoui. A performance comparison of measurement matrices in compressive sensing, WILEY International Journal of Communication Systems, 2018; 31,e3567, pp. 1-18.

o   Abo‐Zahhad MM, Hussein AI, Mohamed AM. Compressive sensing algorithms for signal processing applications: a survey. International J of Communications, Network and System Sciences. 2015;8(6):197‐216.

o   Ji S, Xue Y, Carin L. Bayesian compressive sensing. J IEEE Transactions on Signal Processing. 2008;56(6):2346‐2356.

o   Babacan SD, Molina R, Katsaggelos AK. Bayesian compressive sensing using Laplace priors. J IEEE Trans Image Process. 2010;19(1):53‐63.

o   Baron D, Sarvotham S, Baraniuk RG. Bayesian compressive sensing via belief propagation. J IEEE Trans Signal Process. 2010;58(1): 269‐280.

o   Tropp JA, Gilbert AC. Signal recovery from random measurements via orthogonal matching pursuit. J IEEE trans Info Theory. 2007;53(12):4655‐4666

o   Needell D, Tropp JA. CoSaMP: iterative signal recovery from incomplete and inaccurate samples. J Appl Comput Harmon Anal. 2009;26(3):301‐321

o   Y. Arjoune, N. Kaabouch, H. El Ghazi, and A. Tamtaoui. Compressive sensing: A performance comparison of recovery algorithm, Computing and Communication Workshop and Conference, 2017, pp.1-7. DOI 10.1109/CCWC.2017.7868430

o   The authors need to improve the literature review. The most recent reference cited by the authors goes back to 2016. Please acknowledge the efforts made after 2016 in this context.

6.     How is the multipath modeled in indoor environments?

7.     How many measurements are performed? Which measurement matrix if any was used?

8.     Is one experiment enough to draw conclusions such as the spatial diversity improve RTI systems?

9.     Future research works should be discussed in the conclusion

10.  The authors should discuss the importance of the metrics used for the evaluation and if possible, additional metrics should be included.

Author Response

We would like to thank the reviewer for the valuable comments and suggestions which helped to improve the quality of the manuscript.

Please see the attachment for point-by-point responses.

Reviewer 2 Report

The paper proposes an alternative mechanism for Radio Tomographic Imaging (RTI) that outperforms the nowadays state of the art in accuracy for indoor localization.

The paper is well written and process is acceptably well explained. However, there are some items that, I hope, the authors find useful to improve their paper.

+ I miss clarification on how and why the experiment is performed: why those dimensions? Why this number of links? 

+ After equation 11, I suggest to withdraw the word "Obviously" to avoid claims.

+ Line 161-164: I do not understand what authors mean with this paragraph. What do authors cluster? How is this clustering related with multiple targets? Is there any consideration of losses due to other targets?

+ Some introductory text is missed between section 6 and subsection 6.1.

+ Line 173: "orde" --> "order"

+ Line 181: "channal" --> "channel"

+ Line 198 and Table 2: Some clarification is missed regarding what exactly authors mean with the dB as quality indicators of RTI. 

+ In tables 1 to 4, it would help to add the relative values to compare between the different methods.

+ Line 212: 10MHz --> 10 MHz

+ I miss also some questions to be answered, like the range of application of the method. Would it be useful for big spaces or only for small rooms?

Author Response

(The authors gave the same response as above.)

Reviewer 3 Report

The authors address an important topic. I did not find relevant theoretical errors in the paper, but the writing needs a careful revision.

The authors should compare their techniques with other positioning techniques (both based on the RSS and/or angle-of-arrival/angle-of-departure information), such as the ones proposed by Tomic.

The section with performance results needs particular attention.

Author Response

(The authors gave the same response as above.)

Reviewer 4 Report

The paper involved radio tomographic imaging with spatial diversity based on compressive sensing. It is well organized and is novel. My comments are as follows.

1. Please give the reason of selecting CS for radio tomographic imaging in introduction.

2. Some latest works on CS based on radio tomographic should be involved in introduction such as Device-free localisation with wireless networks based on compressive sensing, Secure Wireless Communications Based on Compressive Sensing: A Survey, etc.

3. The language needs be further improved.

Author Response

We would like to thank the reviewer for the valuable comments and suggestions which helped to improve the quality of the manuscript.

Please see the attachment for point-by-point responses.

Round  2

Reviewer 3 Report

The authors improved the paper, but a careful revision of the writing is still still recommendable.